# Multiple Functions of Hepatitis E Virus ORF3

**DOI:** 10.3390/microorganisms12071405

**Published:** 2024-07-11

**Authors:** Fengyuan Jiao, Yu Zhao, Gengxu Zhou, Chi Meng, Lingjie Wang, Shengping Wu, Jixiang Li, Liting Cao, Bo Zhou, Yichen Luo, Hanwei Jiao

**Affiliations:** 1The College of Veterinary Medicine, Southwest University, Chongqing 402460, China; senrendipity@email.swu.edu.cn (F.J.); zgx973589243@email.swu.edu.cn (G.Z.); mengchi@email.swu.edu.cn (C.M.); guolicheng666@email.swu.edu.cn (L.W.); chemie@email.swu.edu.cn (S.W.); swu_lucky@163.com (J.L.); caoliting@swu.edu.cn (L.C.); 2Ministry of Agriculture and Rural Affairs Key Laboratory of Crop Genitic Resources and Germplasm Innovation in Karst Region, Institute of Animal Husbandry and Veterinary Medicine of Guizhou Academy of Agricultural Science, Guiyang 550005, China; zhaoyu@gzsnky.wecom.work; 3Veterinary Research Institute, Chinese Academy of Agricultural Sciences, Yujinxiang Street 573, Changchun 130102, China; hottank3210@126.com

**Keywords:** HE, HEV, ORF3

## Abstract

Hepatitis E (Hepatitis E, HE) is an acute and chronic infectious hepatitis caused by hepatitis E virus (Hepatitis E Virus, HEV) infection, which is responsible for most acute hepatitis in the world and is a significant public health problem. The pathogen, HEV, has three Open Reading Frames (ORFs) ORF1, ORF2, and ORF3, each of which has a different function. Most of the current research is focused on ORF1 and ORF2, while the research on ORF3 is still relatively small. To provide more ideas for the study of HEV pathogenesis and the prevention and treatment of HE, this paper reviews the effects of ORF3 on the ERK pathway, growth factors, immune response, and virus release.

## 1. Hepatitis E and Its Pathogens

### 1.1. Hepatitis E

HE, an infectious hepatitis caused by the hepatitis E virus (HEV), is one of the most common causes of acute hepatitis in the world [1], causing more than 50% of acute hepatitis in the countries where it develops [2]. Hepatitis E is seen everywhere but occurs most frequently in East and South Asia and is mainly transmitted by the fecal–oral route caused by the fecal contamination of drinking water. The World Health Organization reported in 2017 that the number of people newly infected with HEV can be as high as 20 million per year globally, of which about 3.3 million show symptoms, resulting in about 44,000 deaths. China is a HEV-endemic country [3], and the active prevention, diagnosis, and treatment of hepatitis E has been practiced in China since its discovery. China is the first country to develop and approve a vaccine against the hepatitis E virus, but this vaccine has some limitations, targeting only genotype 1 [4].

### 1.2. Hepatitis E Virus

Hepatitis E virus (HEV) is a small, quasi-enveloped, single-stranded positive-sense RNA virus belonging to the Hepeviridae family [3]. Depending on the internal structure of the virus, it is classified as lax or compact, where the lax type contains a transparent charge region but does not contain complete viral genes and the compact type contains complete viral genes [5]. HEV is similar to the Hepatitis A virus (HAV). It is hypothesized that both viruses are released from cells into the bloodstream as enveloped viruses, which allows them to evade clearance by the immune system in the body until they pass through the intestinal tract and are excreted from the body, where they lose their envelopes to infect the next susceptible host cell.

At present, the hepatitis E virus can currently be divided into eight genotypes, but the common ones are mainly four genotypes: HEV1, HEV2, HEV3, and HEV4. Only HEV3 and HEV4 have animal commonality among these four genotypes and can infect humans and animals [6,7]. HEV1 and HEV2 are mainly transmitted through contaminated water sources, have not been found to have any animal origins, are thought to infect only humans, and are the main hepatitis E pathotypes [8]. They are widespread and highly endemic, especially in Asia, Africa, and South America. HEV3 and HEV4 are zoonotic, infecting domestic pigs, wild boars, rabbits, and other animals and then spreading to humans. HEV3 and HEV4 have become endemic worldwide, mostly in developing countries in Southeast Asia and Latin America.

The amino acid sequences of the ORF3 of different genotypes of hepatitis E were analyzed for homology using MEGA. The results showed that HEV2 and HEV1 had the highest homology and the least divergence of ORF3, while the highest divergence was found in HEV5 and HEV7. The results are shown in the Figure 1 below.

HEV is a spherical particle with a spiky surface and icosahedral symmetry, with a genome length of 7.2 kb, a single-stranded positive-stranded RNA with a cap structure at the 5′ end (m7GpppN), and a Ploy A tail at the 3′ end, and it contains three ORFs [9,10]. HEV ORF1 mainly encodes non-structural proteins associated with the viral genome and is located at the 5´ end of the viral genome, which is about 5 kb long.

A cap structure exists at the 5′ end of ORF1, and the presence of this structure predicts that HEV ORF1 has a role in maintaining the infectivity and activity of the virus. HEV ORF1 is the largest open reading frame encoding the viral replication machinery. It contains many enzymatic regions, including enzymes such as methyltransferases, cysteine proteases, RNA polymerases, and RNA helicases, as well as X, Y, and “hypervariable (HVR)” structural domains of currently unknown function, which provide the necessary conditions for viral replication [11,12].

HEV ORF2 is located at the 3′ end of the viral genome (5147–7129 nt) and is about 2 kb long, with a region of overlap with HEV ORF3. The overlap region is highly conserved and encodes a viral capsid protein of about 660 amino acids in length, which is translated into a capsid protein from a subgenome of 2.2 kb and which has three critical regions, including the S (amino-terminal capsid structural domain), M (intermediate structural domain), and P (carboxy-terminal structural region). These three regions are sequentially connected [9]. As a capsid protein, ORF2 protects the integrity of the viral genome and participates in many important physiological activities such as viral particle assembly, infection–host cell interaction, and immunogenicity [7].

HEV ORF3 overlaps with the N-terminus of ORF2, and the overlap is about 700 nt. The total length of ORF3 varies depending on the HEV genotype and can encode small multifunctional phosphoproteins [13,14]. ORF3, although the most diminutive open reading frame in the HEV genome, contains recognition sequences for various protein kinases, which are thought to play critical roles in signaling and virulence factor release. All three ORFs of HEV regulate many cell signaling pathways and inhibit the host immune response to promote the survival of infected cells. Although some progress has been made in the study of the biological functions of ORF3 in recent years, the study of ORF3 has been relatively superficial compared to that of ORF1 and ORF2, and there are still many questions to be elucidated.

## 2. Structure of HEV ORF3

### 2.1. Structure of ORF3

ORF3 is the smallest ORF in the hepatitis E virus genome, and there are some differences in the size of HEV ORF3 according to virus strains, although it is generally 342 bp~369 bp and overlaps with HEV ORF2. It has been shown that ORF3 is generally localized in early and recirculating endosomes and multivesicular bodies (MVBs) [15,16]. Depending on the HEV genotype encoding 112-, 113-, or 114-amino acid proteins with predicted molecular weights of 12 kDa to 13 kDa, HEV pORF3 is mainly composed of two proline-rich regions, P1 and P2, with two N-terminal hydrophobic regions, D1 and D2. Among them, D1 is rich in cysteine residues and is required for the binding of ORF3 Protein (pORF3) to the cytoskeleton to bind microtubules and mitogen-activated protein kinase (MAPK) [17,18,19]. The P1 region contains two overlapping motifs with homology to kinase; the PMSP module phosphorylated by MAPK and the SPLR, which has loose homology to the SPKK module phosphorylated by cyclin-dependent kinase 1 (CDK1). Two overlapping PXXP (P for proline, X for any amino acid) structural domains capable of binding to the cytosolic protein subunit Src homology 3 domain (SH3) were identified in the P2 region, which has been reported in many viral and cellular proteins involved in signal transduction [20]. They enable pORF3 to interact with various cellular proteins containing the SH3 structural domain [21]. Based on these results, HEV ORF3 may have a signal transduction role in optimizing virus infection and cell replication. Numerous studies have shown that HEV pORF3 interacts with host cell proteins to create a favorable replicative environment for the virus and plays a vital role in the pathogenesis of HEV.

### 2.2. pORF3 of HEV

HEV pORF3 is a phosphorylated protein consisting of 113 or 114 amino acids modified on a single serine residue (Ser-80) by cellular mitogen-activated protein kinase (MAPK), with a relative molecular mass of 13 kDa, multiple functions, and the ability to induce a robust immune response [22,23]. There are two highly hydrophobic regions in the N-terminal positions 16~32 aa and 37~62 aa of pORF3, which may be involved in the process of pORF3’s self-association to form dimers or oligomers. This self-association has specific effects on the function of pORF3 itself [23]. In contrast, Jérôme Gouttenoire et al. showed that the N-terminal end of pORF3 is cysteine-rich, palmitoylated, and decisive for membrane binding and oligomerization [24]. Palmitoylation regulates protein interactions [25]. The palmitoylation of HEV pORF3 determines its membrane binding, subcellular localization, and function, and it may also have a role in stabilizing viral proteins. In addition, the cysteine residue of pORF3 is involved in palmitoylation, severely affecting the secretion of infectious HEV when mutated. The structure is shown in Figure 2.

## 3. Functions of HEV ORF3

### 3.1. Maintains ERK Activity

MAPK, a group of protein serine/threonine kinases, is an essential intracellular signaling regulatory pathway that plays a vital role in the regulation of gene expression and cytoplasmic functional activities. Through the three pathways of MAPK, extracellular regulated protein kinases (ERK), p38 mitogen-activated protein kinase (p38 MAPK), and c-Jun N-terminal kinase (JNK) can play a role in regulating apoptosis, proliferation, and differentiation signaling. Among other things, the activation of the ERK pathway in cells is thought to generate survival and proliferation signals. pORF3 was found to bind to Pyst1, a specific phosphatase in the ERK pathway, through its N-terminal D1 structural domain [21]. The presence of Pyst1 regulates the dephosphorylation of Thr and Tyr in the ERK-activating sequence TXY, thereby inhibiting ERK pathway activity. Therefore, the binding of pORF3 to Pyst1 is required to mediate ERK activation, a process that prevents conformational changes in the phosphatase and maintains ERK activity [20]. Meanwhile, pORF3 can bind to MPK3, inhibiting MPK3 binding to ERK and avoiding ERK inactivation due to the dephosphorylation of MPK3. Their specific linkage is shown in Figure 3.

### 3.2. Inhibition of Growth Factor Transport

When host cells express HEV pORF3, it attenuates mitochondrial death signaling in the cells and delays the transport of activated epidermal growth factor receptor (EGFR) [15,26]. Recently, it has been found to have the same effect on c-Met, the receptor for hepatocyte growth factor (HGF). This action depends on the ability of HEV pORF3 to interact with the multidomain bridging protein CIN85 (Cbl-interacting protein of 85 kDa, CIN85), which HEV pORF3 may bind to via its proline-rich P2 structural domain and CIN85, which contains the SH3 structural domain. It competes with the growth factor receptor-Cbl-CIN85 complex, reducing the translocation of activated growth factors and their degradation [27]. This condition will prolong endothelial growth factor signaling and promote cell survival and proliferation.

### 3.3. Suppression of the Immune Response

The phosphorylation of the phosphorylated form of signal transducer and activator of transcription 3 (STAT3) requires endocytosis and intracellular transport by the cell growth factor receptor [28]. In nuclei expressing HEV pORF3, the level of phosphorylated STAT3 (pSTAT3) was reduced due to its effect on epidermal growth factor receptor (EGFR) transport, which led to a decreased expression of proteins involved in the acute-phase response and an attenuated signaling of inflammatory factors. In addition to this, ORF3 also has an inhibitory effect on nuclear factor kappa-B (NF-κB) signaling [29]. Physically, the mutual binding of RIP1 and the β-interferon TIR structural domain interface protein TRIF promotes the activation of TAKI, a member of the MAPK family of proteins, which is capable of phosphorylating IKKβ (inhibitor of nuclear factor kappa-B kinase) and thereby increasing its enzymatic activity [30]. In contrast, the NF-κB protein is usually inactivated by forming dimers from p65 and p50, which bind to the inhibitory protein IkB to form a trimeric complex. Phosphorylated IKKβ will then catalyze the phosphorylation of IκBα and dissociate it from the trimer, ultimately leading to NF-κB activation. Within this, on the one hand, the tumor necrosis factor receptor-associated death domain protein (TRADD) promotes the K63 ubiquitination of receptor-interacting protein kinase 1 (RIP1), while ORF3 degrades TRADD and directly inhibits K63 ubiquitination of RIP1; on the other hand, ORF3 also inhibits the activation of IKKβ and IκBα. Therefore, HEV ORF3 also exerts an inhibitory effect on the NF-κB pathway. The specific relationship is shown in Figure 4.

Meanwhile, in HEV pORF3-expressing cells, the conserved PSAP modality in the P2 structural domain binds to pORF3 under the mediation of tumor susceptibility gene 101 (Tsg101) protein, leading to an increase in α-1-microglobulin (α-MG) secretion [31,32]. Since α-MG is an immunosuppressive protein, the idea that α-MG can protect cells infected by HEV was proposed. One trial reported significantly higher urinary alpha-MG levels in hepatitis E patients compared to acute hepatitis B patients and healthy controls [33], which also laterally validated this assumption. It has also been reported that HEV pORF3 binds to heme through its D2 structural domain [29,34] and plays a vital role in iron ion metabolism, aiding viral infection by affecting cellular iron ion homeostasis.

### 3.4. Interacts with pORF2

Graff et al. found that ORF2 protein production is influenced by the nucleotide sequence at the 5 end of ORF3 [35]. Although pORF3 can interact with pORF2 and the mechanism of interaction is not clear, HEVs lacking the ORF3 gene are unable to infect rhesus monkeys. ORF2 mainly encodes proteins of the coat, which leads to the speculation that ORF3 can regulate the assembly of proteins of the coat via ORF2. In addition, the phosphorylation of ORF3 Ser80 greatly influences the interaction between ORF2 and ORF3. If Ser-80 is mutated and dephosphorylated, the binding of pORF3 to pORF2 disappears, and the results indicate that the interaction between pORF3 and pORF2 is dependent on the phosphorylation of the Ser-80 site [36]. PantevaM et al. found that the binding of pORF3 to non-glycosylated pORF2 accelerated host cell death and the release of mature hepatitis E virus particles [37].

### 3.5. Participation in Virus Release

The ORF3 protein, located on the surface of HEV, is a functional viral channel protein that mediates viral release via the exosome pathway by targeting the viral capsid to the endosomal system, where endosomal targeting is initiated by covering the viral capsid with Porf3 [37,38]. Yamada et al. [39] found that the infectious cDNA clone pJE03-1760 F/wt of the orf3 deletion mutant could replicate efficiently in PLC/PRF/5 and A549 cells. However, the amount of virus detected in the culture supernatants of ORF3-deficient mutant-infected cells was only 1% of that of the wild-type clone-infected cells, suggesting that ORF3 plays an irreplaceable role in virus release. The results of an intrahepatic inoculation assay in rhesus monkeys showed that pORF3 is required for HEV infection and functions as a toxic auxiliary protein. The replication of genotype 1 HEV in Caco-2 and Huh-7 cells is dependent on pORF3 expression [40], and HEV viruses are present in feces as naked spherical particles (nHEV) and in circulating blood as quasi-enveloped particles encapsulated in a lipid layer (eHEV) [41]. These results show that HEV pORF3 can be present on the surface of the virus in association with lipids, and the periplasm formed is shed as the virus passes through the intestinal system. Based on this phenomenon, it is hypothesized that there is a link between the formation of the periplasm and the release of the virus. Kentaro Yamada et al. also hypothesized experimentally that HEV is released from infected cells as lipid-associated viral particles. Then, pORF3 is shed in bile and dissociated from the viral particles [39].

Some RNA viruses also have proline-rich sequences necessary for viral outgrowth that are located in structural domains at the rear of the virus, including the P(S/T)AP and PPXY (X represents any amino acid) motifs [29]. The formation of the viral surface periplasm by pORF3 requires an intact PSAP module in the P2 structural domain [32]. It may be related to the mediation of the Tsg101 protein, which has previously been shown to bind to pORF3. The PSAP motif of HEV pORF3 is a functional structural domain for the release of viral particles, and an intact PSAP motif is critical for the efflux of lipid- and pORF3-associated HEV particles. ORF3 has at least two PASP motifs, and both may be involved in the exocytosis of viral particles, with at least one required to form and release membrane-bound HEV particles with pORF3 on the surface [32]. Although HEV is a non-enveloped virus, the intracellular localization of HEV pORF3 and the requirement of the periplasm for an intact PSAP module in the P2 structural domain suggest that HEV can be released from cells via the VPS pathway. It has also been reported that HEV pORF3 possesses ion channel activity required for the release of infectious viruses, which is essential for the formation of ion channels because it is a transmembrane protein located in the endoplasmic reticulum (ER) membrane and can form multimeric complexes [14]. The mechanism by which ORF3 helps virus release is shown in Figure 5.

### 3.6. Affects Lipid Metabolism

Currently, with the in-depth study of HEV ORF3, it has been found that the expression of HEV ORF3 in the overall membrane proteins and basement membrane proteins of the host cell is affected by ORF3 during the viral infection of the host cell, which may lead to altered lipid metabolism processes [42]. The researchers found HEV pORF3 present in the apical membrane of released viruses in depolarized Caco-2 cells, and the phenomenon may be lipid-related. Moreover, it was found that the hepatitis E virus produced in cell culture had a lipid envelope. However, the virus found in feces was a naked nucleocapsid with no lipid layer or surface expression of the ORF3 protein [3,43]. In contrast, the results of Takahashi et al. showed the presence of the ORF3 protein bound to lipids on the surface of HEVs in serum [44]. After transcriptome sequencing, it was found that APOC3, SCARA3, DKK1, and other genes related to lipid metabolism were dysregulated after the stable expression of HEV pORF3 in HepG2 cells [42]. It was hypothesized that the expression of SHEV pORF3 might affect the normal lipid metabolism of the host cells due to this result.

### 3.7. Helping the Virus to Replicate

Several strategies are generally employed by viruses to evade the host’s immune system or to establish an immunosuppressive environment after entering the host, among which is the interferon system, which plays a vital role in the host’s antiviral response. In interferon induction, HEV ORF3 plays both positive and negative roles. On the one hand, the HEV ORF3 protein mainly activates RIG-1 by prolonging the half-life of retinoic acid-inducible gene I (RIG-I) protein and interacting with the N-terminus of RIG-1, thus enhancing IFN induction. On the other hand, HEV ORF3 blocks the phosphorylation of signal transducer and activator of transcription 1 (STAT1) to inhibit Janus kinase/STAT signaling. The JAK-STAT signaling pathway plays a crucial role in interferon-induced antiviral responses [13,22]. It was found that HEV ORF3 could reduce the expression of CD14 and CD64 by inhibiting the JAK1\STAT1 signaling pathway, which significantly impaired the phagocytosis of phagocytes, while the down-regulation of CD14 and CD64 could be reversed by interferon [45]. HEV ORF3 also significantly impaired endogenous type I interferon production by down-regulating TLR3 and TLR7, which was achieved by impairing a variety of corresponding signaling pathways, including NF-κB, JAK\STAT, and JNK\MAPK.

Moreover, ORF3 inhibits the secretion of inflammatory molecules from THP1 macrophages by inducing tumor necrosis factor-α (TNF**-α**) to inhibit the activation of the NF-κB pathway, leading to a decrease in the level of expression of inflammatory response genes and the creation of a favorable environment for viral replication [46]. Interferon induces STAT1 phosphorylation, and ORF3 can form a complex with STAT1 to inhibit STAT1 phosphorylation; the inhibition of STAT1 phosphorylation results in the inhibition of the expression of the antiviral genes PKR (double-stranded RNA-activated protein kinase), 2,5-oligoadenylate synthetase (2,5-OAS), and myxomatosis viral resistance protein A (MxA) and the subsequent evasion of host attack [22]. Therefore, ORF3 can aid viral replication and evade host immune clearance by inhibiting endogenous interferon secretion.

## 4. Conclusions and Outlook

In summary, HEV ORF3 can promote cell survival and proliferation in HEV infections by regulating the ERK pathway and the transport of growth factors to regulate apoptosis and death signaling. ORF3 also inhibits innate host responses by attenuating acute-phase responses and increasing the secretion of immunosuppressive factors (e.g., α-1-microglobulin). It may also regulate cellular energetic homeostasis by affecting host cell iron metabolism versus lipid metabolism, facilitating viral survival and cell replication.

Hepatitis E is a critical public health problem, and HEV vaccine research is gradually becoming a hotspot. The only HEV prophylactic vaccine, Heconlin, is currently only licensed in China, and the lack of an efficient in vitro culture system limits our knowledge related to the use of HEV viruses for vaccine development. The main direction of vaccine development is genetically engineered subunit vaccines [47,48]. Since ORF2 encodes the virus’s protein capsid and has a strong antigenic epitope region with good immunogenicity, most of the current screening for antigenic epitopes is based on HEV pORF2. Pezzoni et al. showed that the region from 394 aa to 608 aa of the ORF2 protein, which corresponds to the fifth and sixth antigenic regions targeted by monoclonal antibodies (MAbs), is the most immunogenic and contains conformational and linear epitopes [7] as well as three glycosylation sites necessary for the formation of infectious particles [14]. Whether ORF3 possesses a neutralizing epitope is unclear. However, studies have shown that the C-terminus of the ORF3 protein, which is present in human, porcine, and avian HEV, is highly immunogenic and triggers a humoral response in the host [49] Yan et al. prepared three ORF3 protein subunit vaccines via prokaryotic expression. They showed that these three ORF3 subunit vaccines significantly blocked viral shedding in the feces after HEV infection and had good immunogenicity [48]. CD4 cells play an essential role in virus infection, and Csernalabics et al. found that HEV-specific CD4 T-cells mainly target ORF2-derived capsids [50]. The ORF2 protein mainly induces an immune response to produce IgG antibodies, whereas the ORF3 protein induces the production of IgM antibodies [48]. Would a vaccine combining ORF2 with ORF3 be more effective? The current revelation and elucidation of these mechanisms of action still need to be more comprehensive and specific, so more in-depth research on HEV ORF3 is needed to provide a more theoretical basis and direction for the prevention and treatment of HE.

## Figures and Tables

**Figure 1 microorganisms-12-01405-f001:**
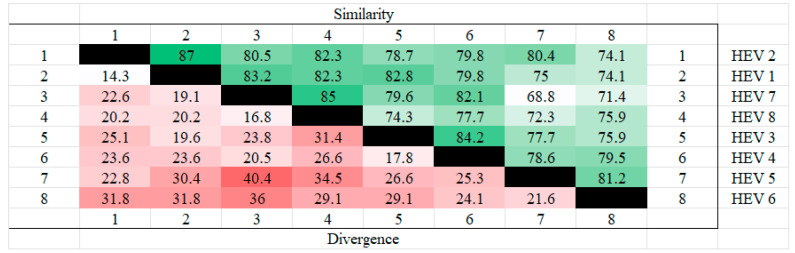
Homozygosity analysis of ORF3 in different genotypes. HEV 1(AAA45735.1), HEV2(AAA45731.1), HEV3(BAC44898.1), HEV 4(BAE02702.1), HEV5(BAJ77117.1), HEV 6(BAO31622.1), HEV7(AHY61297.1), HEV8(AOR52321.1). From the figure, it can be seen that HEV1 and HEV2 have the highest ORF3 homology and HEV7 and HEV8 also have high homology, whereas the highest dissimilarity is found in HEV5 and HEV7, followed by HEV6 and HEV7, which also have high dissimilarity.

**Figure 2 microorganisms-12-01405-f002:**
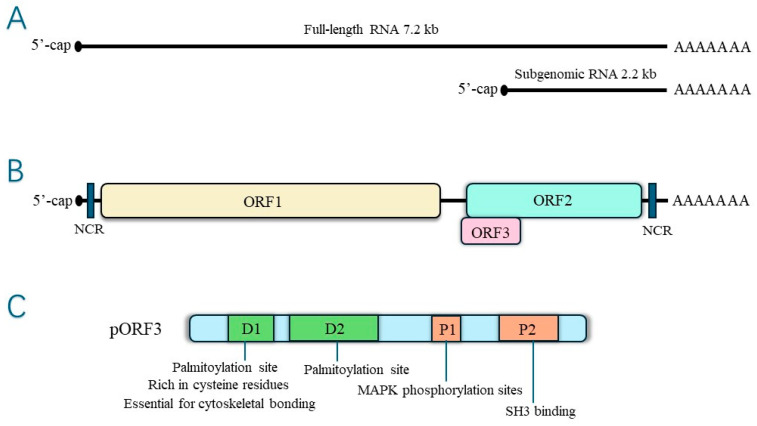
(**A**)The hepatitis E virus genome RNA—genomic RNA and bicistronic subgenomic RNA. (**B**) Open reading frames. (**C**) Structure of ORF3 protein.

**Figure 3 microorganisms-12-01405-f003:**
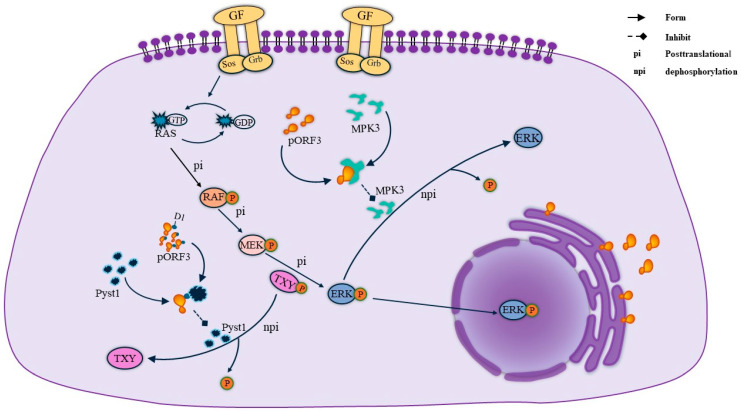
ORF3 maintains ERK activity. On the one hand, the ORF3 protein maintains ERK activity by binding to pyst1, thereby inhibiting the dephosphorylation of TXY. On the other hand, ORF3 directly binds to MPK3 to form a complex, which reduces ERK because of the dephosphorylation of MPK3 and thus comes to maintain ERK activity.

**Figure 4 microorganisms-12-01405-f004:**
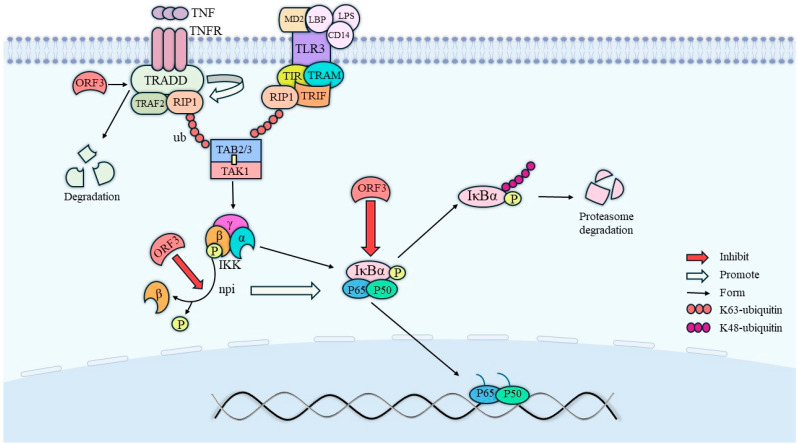
Pathways of NF-κB activity inhibition by ORF3. ORF3 directly promotes the degradation of TRADD to inhibit NF-κB activity, on the one hand, and inhibits NF-κB activity by directly inhibiting the dephosphorylation of IKKβ, on the other hand. In addition, IκBα activation needs to be facilitated by the dephosphorylation of IKKβ, so ORF3 not only directly inhibits IκBα activation, but also indirectly inhibits IκBα activation by inhibiting IKKβ activation.

**Figure 5 microorganisms-12-01405-f005:**
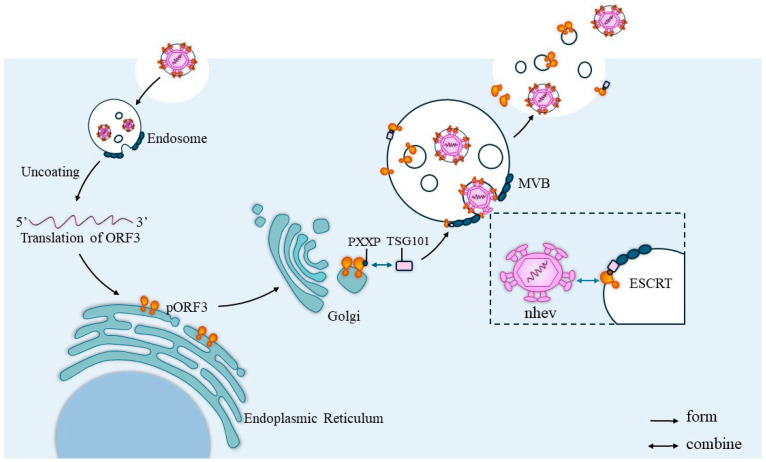
ORF3 helps virus release. After the HEV invades the cell, it will first localize into the endosomes and, after a series of activities, the virus will delocalize for translation and the PXXP sequence of the ORF3 protein will bind to the TSG101 protein in the ESCRT system and translocate to the ESCRT. Then, the naked virus particles will interact with the translocated pORF3, which will cover the surface of the viral capsid and initiate virus release via the exosome pathway.

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
