# Peer review of "Multiple Functions of Hepatitis E Virus ORF3"

_microorganisms, 2024, doi:10.3390/microorganisms12071405_

Round 1

Reviewer 1 Report

Comments and Suggestions for Authors

The review "ORF3 of Swine Hepatitis E Virus (HEV)" summarizes the main role and interaction of ORF3 protein. This is an interesting and unexplored topic, however, there are already some reviews on this subject. Authors should include more novel information in order to present a more relevant article.

1.The title makes reference to swine ORF3, but is there a clear difference between swine and human ORF3? I suggest to rephrase the title.

2.In order to provide more data on this protein, I suggest to include a thorough structural and sequence analysis comparing between strains from different genotypes.

3.Figure 1 is not very clear and the pathway interaction could be more detailed.

4.I suggest to include a figure of the genome structure and main domains of ORF3.

5.The abbreviation HE in the abstract is not define.

6.The HEV vaccine is only available for genotype 1, please clarify this in the text.

7.HEV is currently classified in Hepeviridae family according to ICTV, please revise. 

8.Line 67 has a typo.

Comments on the Quality of English Language

Please revise and rephrase some long sentences which can be confusing.

Author Response

  1. The title makes reference to swine ORF3, but is there a clear difference between swine and human ORF3? I suggest to rephrase the title.

Thank you for your suggestion, as the species is not very strong in this article, with no clear distinction between humans and pigs, the title of the article has now been changed to "Multiple functions of Hepatitis E virus ORF3."

  1. In order to provide more data on this protein, I suggest to include a thorough structural and sequence analysis comparing between strains from different genotypes.

In response to the suggestions you gave, we analyzed the homozygosity of ORF3 in different genotypes separately, and the results are as follows

3.Figure 1 is not very clear and the pathway interaction could be more detailed.

Figure 1 has been modified:

4.I suggest to include a figure of the genome structure and main domains of ORF3.

We added a figure of the HEV viral genome and the structure of PORF3 as below.

5.The abbreviation HE in the abstract is not define.

Hepatitis E in the abstract has been revised to read as follows: Hepatitis E (Hepatitis E ,HE)

6.The HEV vaccine is only available for genotype 1, please clarify this in the text.

The text has been changed to’Only China is the first country to develop and approve a vaccine against the hepatitis E virus, but this vaccine has some limitations, targeting only genotype 1.[4]’

7.HEV is currently classified in Hepeviridae family according to ICTV, please revise. 

The text has been changed to’ Hepatitis E virus (HEV) is a small, quasi-enveloped, single-stranded positive-sense RNA virus belonging to the Hepeviridae family [3].’

8.Line 67 has a typo

The error has been corrected.

Comments on the Quality of English Language

Please revise and rephrase some long sentences which can be confusing.

We apologize for the poor language of our manuscript. We worked on the manuscript for a long time and the repeated addition and removal of sentences and sections obviously led to poor readability. We've overhauled the grammar of this post in hopes of increasing readability.

Reviewer 2 Report

Comments and Suggestions for Authors

The manuscript under evaluation is a review about the Open Reading Frame in Swine Hepatitis E virus.

General comment. The topic of the study is of narrow interest. Whilst the manuscript is surely within the remit of the journal, I find that the length of the manuscript is rather excessive. This is corroborated by the inclusion of only 46 references, which is really reasonable, because indeed the topic is narrow. Hence, I suggest to the authors to reduce the length of the revised manuscript.

The extent of the topic of the review does not justify the length of the manuscript. This is a significant problem, which clearly requires extensive revision of the submission.

Moreover, the authors must indicate the advantages of their review in comparison to other relevant reviews published previously on the same virus.

The authors should provide a brief section to explain the methodology through which they chose the references for inclusion in the manuscript. For example, which databases did they use to search for references? Which keywords did they employ? Etc.

The authors should also include a separate sub-section to present some data regarding ORF in other viruses and to compare with Swine Hepatitis E virus.

Finally, the manuscript requires a general revision of the English language.

Comments on the Quality of English Language

Finally, the manuscript requires a general revision of the English language.

Author Response

The extent of the topic of the review does not justify the length of the manuscript. This is a significant problem, which clearly requires extensive revision of the submission.

 Thank you for your suggestion; based on your suggestion, we have reorganized the article structure, deleted some content about ORF1 and ORF2, and added some research on the ORF3 vaccine, focusing more on ORF3.

Moreover, the authors must indicate the advantages of their review in comparison to other relevant reviews published previously on the same virus.

This article provides an overview of the multiple functions of ORF3 inside HEV viruses, including its effects on the ERK pathway, growth factors, immune response, and virus release. This may provide more ideas for the study of HEV pathogenesis and how to prevent and control HE

 The authors should provide a brief section to explain the methodology through which they chose the references for inclusion in the manuscript. For example, which databases did they use to search for references? Which keywords did they employ? Etc.

I chose my references by using a keyword search inside Pubmed in the NCBI database, searching for HEV ORF3, HEV ORF3&vaccine, HEV ORF3&MAPK, etc.

 The authors should also include a separate sub-section to present some data regarding ORF in other viruses and to compare with Swine Hepatitis E virus.

In response to the suggestions you gave, we analyzed the homozygosity of ORF3 in different genotypes separately, and the results are as follows

 Finally, the manuscript requires a general revision of the English language.

In response to the suggestions you've given, we've overhauled the grammar of the article, so please be sure to point out if there's anything else that's not right.

Comments on the Quality of English Language

Finally, the manuscript requires a general revision of the English language.

Round 2

Reviewer 1 Report

Comments and Suggestions for Authors

The manuscript underwent beneficial modifications. I still have some minor comments. 

1.L. 59 The identity matrix is based on aminoacid sequences not nucleotide, please check. 

2.The Figure 1 is missing the figure caption, a more detail description should be included.

3.The software employed for the matrix construction should be described.

4.L. 333 is not clear why the HEV detection method is mentioned among ORF2 and ORF3 vaccine discussion.

5.I suggest to rephrase section 1 title: "Hepatitis E and its pathogens"

Author Response

1.L. 59 The identity matrix is based on aminoacid sequences not nucleotide, please check. 

Thank you for your attentiveness, we have modified this sentence to read as follows:

The amino acid sequences of the ORF3 of different genotypes of hepatitis E were analyzed for homology using MEGA.

2.The Figure 1 is missing the figure caption, a more detail description should be included.

In response to your suggestion, we have added some clarification to Figure 1.

Figure 1: Homozygosity analysis of ORF3 in different genotypes. HEV 1(AAA45735.1),HEV2(AAA45731.1),HEV3(BAC44898.1),HEV 4(BAE02702.1),HEV5(BAJ77117.1),HEV 6(BAO31622.1),HEV7(AHY61297.1),HEV8(AOR52321.1). From the figure, it can be seen that HEV1 and HEV2 have the highest ORF3 homology, and HEV7 and HEV8 also have high homology, whereas the highest dissimilarity is found in HEV5 and HEV7, followed by HEV6 and HEV7 also having high dissimilarity. 

3.The software employed for the matrix construction should be described.

Thank you for your suggestion, we have indicated in the text that the software used is MEGA.

The amino acid sequences of the ORF3 of different genotypes of hepatitis E were analyzed for homology using MEGA.

4.L. 333 is not clear why the HEV detection method is mentioned among ORF2 and ORF3 vaccine discussion.

Based on your suggestion, we found that this sentence is not very relevant to the article and is a bit off-topic, so we have now removed this sentence, thank you for your careful review.

5.I suggest to rephrase section 1 title: "Hepatitis E and its pathogens"

Thanks for the suggestion, we've changed the title to:Hepatitis E and its pathogens

Reviewer 2 Report

Comments and Suggestions for Authors

The authors made the corrections suggested and improved the submission in the revised manuscript.

Author Response

Thank you for your decision and constructive comments on my manuscript.